# Ultrasonic-Assisted Extraction, Structural Characteristics, and Antioxidant Activities of Polysaccharides from *Alpinia officinarum* Hance

**DOI:** 10.3390/foods13020333

**Published:** 2024-01-20

**Authors:** Xuejing Jia, Guanghuo Liu, Yun Huang, Zipeng Li, Xiaofei Liu, Zhuo Wang, Rui Li, Bingbing Song, Saiyi Zhong

**Affiliations:** Guangdong Provincial Key Laboratory of Aquatic Products Processing and Safety, Guangdong Provincial Science and Technology Innovation Center for Subtropical Fruit and Vegetable Processing, Guangdong Provincial Engineering Technology Research Center of Seafood, College of Food Science and Technology, Guangdong Ocean University, Zhanjiang 524088, China; jiaxj@gdou.edu.cn (X.J.); 13531035735@163.com (G.L.); woej@stu.gdou.edu.cn (Y.H.); 202111221317@stu.gdou.edu.cn (Z.L.); liuxf169@126.com (X.L.); wangzhuo4132@outlook.com (Z.W.); liruihn@163.com (R.L.); 15891793858@163.com (B.S.)

**Keywords:** *Alpinia officinarum*, polysaccharide, ultrasonic-assisted extraction, physicochemical characteristics, antioxidant activity

## Abstract

*Alpinia officinarum* Hance, a well known agricultural product in the Lei Zhou peninsula, is generally rich in polysaccharides. In order to enhance the use of *A. officinarum* Hance polysaccharides (AOP) in functional food, AOP was extracted using an ultrasonic-assisted extraction method, and the ultrasonic extraction parameters of AOP was optimized. Furthermore, this study investigated the physicochemical and antioxidant activities of AOPs. In addition, the structural properties were preliminarily determined using Fourier-transform infrared spectroscopy (FTIR), high performance size exclusion chromatography, and a Zetasizer. Ultimately, this study explored the mechanism underlying the antioxidant activities of AOP. The results showed that the optimal ultrasonic-assisted extraction parameters were as follows: ultrasonic time, 6 min; ratio of water to material, 12 mL/g; and ultrasonic power, 380 W. Under these conditions, the maximum yield of AOPs was 5.72%, indicating that ultrasonic-assisted extraction technology is suitable for extracting AOPs due to the reduced time and water usage. Additionally, AOPs were purified using graded alcohol precipitation, resulting in three fractions (AOP30, AOP50, and AOP70). AOP30 had the lowest molecular weight of 11.07 kDa and mainly consisted of glucose (89.88%). The half inhibitory concentration (IC_50_) value of AOP30 and AOP70 was lower than that of AOP50 in the ability to scavenge the ABTS radical, while a reverse trend was observed in reducing ferric ions. Notably, the antioxidant activities of AOPs were highly correlated with their polydispersity index (Mw/Mn) and Zeta potential. AOP30, a negatively charged acidic polysaccharide fraction, exhibited electron donating capacities. Additionally, it displayed strong antioxidant abilities through scavenging 2,2′-azinobis-(3-ethylbenzthiazoline-6-sulphonate) (ABTS) radicals and reducing ferric ions. In conclusion, the present study suggests that AOP30 could be developed as an antioxidant ingredient for the food industry.

## 1. Introduction

*Alpinia officinarum* Hance, commonly named Gao-Liang-Jiang, belongs to the Zingiberaceae family and is one of the most known culinary spices in southern China. Its rhizome is 4–10 cm in length and dark brown in color [1]. It has been utilized for culinary purposes in southern China for several hundred years. In rural areas, it is consumed as a dietary product, such as in soup and in porridge; therefore, it has been included in the list of Affinal Drugs and Health Foods published by the China Food and Drug Administration since 2021 [2]. Additionally, phytochemical evidence of *A. officinarum* indicates that p-octopamine, essential oils, and phenylpropanoids are abundant in the rhizomes [3,4,5,6]. These small molecular compounds display versatile biological activities. For instance, alleviated *Helicobacter pylori*-associated gastritis inhibited the growth of common pathogenic bacteria; reduced the secretion of tumor necrosis factor-α, interleukin-1β, and interleukin-8 in lipopolysaccharide-treated murine macrophage J774A1; and regulated the levels of superoxide dismutase and malonaldehyde [7,8,9]. However, the research on biomacromolecules in *A. officinarum* is inferior to the abundant studies on small molecular compounds. It was reported that *A. officinarum* contains 20.25% carbohydrates [5]. A previous study indicated that *A. officinarum* polysaccharides (AOPs) decreased the activity of tyrosinase with an IC_50_ value of 0.315 mg/mL [10]. In addition, AOPs enhanced the proliferation of the murine spleen cells [11]. In order to utilize AOPs, many trials have been conducted. The hot water method was applied to isolate AOPs under the following conditions: extraction temperature, 95 °C; extraction time, 3 h; and ratio of water to material, 43 mL/g [12]. The enzymolysis method was also conducted under the following conditions: ratio of water to material, 24 mL/g; enzymolysis time, 50.5 min [13]; and scavenging of ABTS radicals with an IC_50_ value of 4.33 mg/mL [14]. Furthermore, the microwave-assisted method was performed under the following conditions: microwave time, 20 min; microwave power, 350 W; ratio of water to material, 45 mL/g [15]. The following three findings were accordingly confirmed on the basis of the above extraction methods when extracting AOPs: firstly, a large amount of time was consumed, which failed to meet the standards of food manufacturing; secondly, a great amount of water was required because of a high liquid–solid ratio, which would take considerable time and energy to concentrate the extracting solution. Thirdly, the mechanism underlying the antioxidant activity of AOPs was unclear. Hence, there is a high demand to develop an effective technology to separate AOPs and, at the same time, elucidate the mechanism of antioxidant activities.

Ultrasonication, an energy saving technology consisting of mechanical waves along with extremely short wavelengths with frequencies greater than 16 kHz, is generally performed to break up the cell wall and release active ingredients from the food’s raw material. Ultrasonication shows advantages in accelerating the extraction process, facilitating the dissociation between targeted components and the raw material, and operating under high intensity [16]. Indeed, ultrasonic technology is widely utilized to isolate bioactive polysaccharides from the raw material; for example, when isolating *Hemerocallis citrina* polysaccharides, the ratio of water to material was 25 mL/g, and while extracting *Orchis chusua D. Don* polysaccharides, the ultrasonic time was 50 min [17,18]. This evidence indicates that ultrasonic-assisted extraction is an environmentally friendly technology for isolating polysaccharides. Moreover, due to the function of cavitation that slightly increases the temperature and pressure of the solution, it slowly produces small bubbles among the solution and the raw material [19]. Therefore, it was decided to investigate whether the ultrasonic treatment is beneficial for extracting AOPs. As a result, to reduce the investment in raw material and overcome the abovementioned questions when obtaining AOPs, this present study aimed to optimize the ultrasonic-assisted extraction parameters of AOPs using the Box–Behnken design, analyze structural features, and, importantly, elucidate the underlying mechanism behind the antioxidant activities of AOPs.

## 2. Materials and Methods

### 2.1. Materials and Chemicals

*A. officinarum* Hance rhizomes were purchased from the market of Long tang town (Xuwen county, Zhanjiang, Guangdong province, China). The raw material was disintegrated with a laboratory disintegrator (800Y, Bo ou Hardware Factory, Taizhou, China) into powder, screened using an 80-mesh sieve, and stored in a desiccator to avoid any contact with moisture.

Vitamin C, KBr, and ethanol were purchased from Aladdin Industrial Corporation (Shanghai, China), Sigma-Aldrich Chemical Co., Shanghai, China), and Xi Long Chemical Co., Ltd. (Tianjin, China), respectively; all other used reagents were of analytical grade.

### 2.2. Isolation of Polysaccharides from A. officinarum Hance

#### 2.2.1. Pretreatment of Raw Materials

The powder of *A. officinarum* Hance rhizome was refluxed with anhydrous ethanol three times (each for 2 h) to remove lipids, pigments, and alcohol-soluble constituents. The treated powder was filtered and air dried for further utilization.

#### 2.2.2. Ultrasonic-Assisted Extraction

Ultrasonic-assisted isolation was conducted as per a previous method [20]. Briefly, 2 g of pretreated *A. officinarum* powder was mixed with water and treated with an ultrasonic cleaner (KQ-500DB, Jiangsu Kunshan ultrasonic instrument Co., Ltd., Kunshan, China). The supernatant was obtained after centrifugation for 10 min at 5000 rpm, and the contents of the polysaccharides were determined using the phenol–sulfuric acid method. In addition, the yield of AOPs (%) was calculated using the following equation:

$$Y=\frac{C\times V\times N}{W}\times 100,$$
where *Y* is the yield of AOP, *C* is the concentration of polysaccharides calculated from the calibrated regression equation (μg/mL), *V* is the total volume of extraction solution (mL), *N* is the dilution factor, and *W* is the weight of pretreated *A. officinarum* (μg). 

Finally, the effect of various factors (ultrasonic time, ultrasonic power, and ratio of water to material) on the yield of AOPs was measured.

#### 2.2.3. Optimal Ultrasonic-Assisted Isolation of AOPs

The Box–Behnken design (BBD) was applied to evaluate the effects of ultrasonic parameters on the yield of AOPs. The independent variables included ultrasonic power (A, 380, 475, 560 W), ultrasonic time (B, 4, 5, 6 min), and ratio of water to raw material (C, 8, 10, 12 mL/g). Seventeen trials within five central points were performed based on BBD as shown in Table 1; later, the experimental data of BBD were fitted using the following second-order polynomial model [20]: 

$$Y=\beta_{0}+\sum_{3}^{i=1}\;\beta_{i}X_{i}+\sum_{3}^{i=1}\;\beta_{ii}X_{i}^{2}+\sum_{3}^{i=1}\;\sum_{3}^{j=i+1}\;\beta_{ij}X_{i}X_{j}$$
where *Y* is the predicted yield of AOP; *β*_0_, *β_i_*, *β_ii_*, and *β_ij_* are the regression coefficients for intercept, linearity, square, and interaction, respectively; and *X_i_* and *X_j_* are independent variables (*i* ≠ *j*), respectively.

#### 2.2.4. Purification

AOPs were purified using graded-alcohol precipitation, and three fractions (AOP30, AOP50, AOP70) were collected and freeze dried.

### 2.3. Determination of Physicochemical and Structural Features of AOPs

#### 2.3.1. Analysis of Polysaccharides Content

Polysaccharide content was determined using the phenol–sulfuric acid method with slight modifications [21]. In brief, 0.25 mL AOP solutions (0.25, 0.5, 1, 2, 4, and 8 mg/mL) were mixed with 1.25 mL of sulfuric acid and 0.25 mL of 5% phenol solution. The absorbance was read at 490 nm (Varioskan Flash, Thermo Scientific, Waltham, MA, USA) using D-glucose as the standard.

#### 2.3.2. Analysis of Protein Content

Protein content was determined using the bicinchoninic acid/CuSO_4_ (BCA) method. Briefly, 20 μL of AOP solution was fully mixed with a BCA kit (Beyotime, (Shanghai, China) P0010) for 30 min at 37 °C. Then, the absorbance was read at 562 nm (Varioskan Flash, Thermo Scientific, USA), using D–glucose as the standard.

#### 2.3.3. Analysis of Uronic Acid Content

Uronic acid content was determined using the m-hydroxyl-biphenyl method [22]. In brief, 0.25 mL of AOP solution was mixed with 1.25 mL 0.48% sodium tetraborate (dissolved in concentrated sulfuric acid) and 25 μL of 0.15% m-hydroxyl-biphenyl (dissolved in 0.5% NaOH). Thereafter, the absorbance was read at 525 nm (Varioskan Flash, Thermo Scientific, USA), using galacturonic acid as standard.

#### 2.3.4. Analysis of Fourier-Transform Infrared Spectroscopy

The organic functional groups of AOPs were identified using FTIR spectroscopy and recorded with an infrared spectrometer (Tensor 27, Bruker Optics, Leipzig, Germany) in a range of 500 to 4000 cm^−1^. In brief, a 5 mg sample was mixed with 200 mg of KBr under an infrared lamp, then pressed into a 1 mm disk.

#### 2.3.5. Analysis of Molecular Mass Distribution

A molecular weight of AOPs was dissolved in 0.1 mol/L NaNO_3_ aqueous solution to a final concentration of 1 mg/mL, centrifuged at 12,000 rpm for 10 min, and filtered through a filter with a 0.45 μm pore size. The supernatant was measured using an HPSEC system with a MALLS detector (DAWN HELEOS II laser photometer, Wyatt Technology, Santa Barbara, CA, USA) using the following conditions: temperature, 45 °C; flow rate, 0.6 mL/min; and a differential refractive index detector (Optilab T-rEX, Wyatt Technology Co., Goleta, CA, USA), which was simultaneously connected to obtain the parameters of molecular mass. Technical support was provided by Shanghai Sanshu Biotechnology Co., Ltd. (Shanghai, China).

#### 2.3.6. Analysis of Monosaccharide Composition

Five milligrams of AOP solution were hydrolyzed with 2 mol/L trifluoroacetic acid (TFA) at 121 °C for 2 h in a sealed tube. Then, the hydrolysate was dried with nitrogen gas and methanol to remove excess TFA three times. Subsequently, the residue was dissolved in deionized water and filtered through a 0.22 μm microporous filtering film and analyzed using high–performance anion–exchange chromatography (HPAEC) with a CarboPac PA–20 anion–exchange column of 150 mm × 3 mm, 6.5 μm, Dionex, Sunnyvale, CA, USA), which was equipped with a pulsed amperometric detector (PAD; Dionex ICS 5000+ system) under the following conditions: flow rate, 0.5 mL/min; injection volume, 5 μL; solvent system A: (ddH_2_O), solvent system B: (0.1 mol/L NaOH), solvent system C: (0.1 mol/L NaOH, 0.2 mol/L NaAc); gradient program, volume ratio of solution A, B, C was 95:5:0 at 0 min, 85:5:10 at 26 min, 85:5:10 at 42 min, 60:0:40 at 42.1 min, 60:40:0 at 52 min, 95:5:0 at 52.1 min, and 95:5:0 at 60 min. Technical support was provided by Shanghai Sanshu Biotechnology Co., Ltd. (Shanghai, China).

#### 2.3.7. Analysis of Zeta Potential

The Zeta potential of AOPs was determined using a Malvern Zetasizer Nano-ZSE (Malvern Instruments Ltd., Malvern, UK) according to a previously reported method [23].

### 2.4. Determination of Antioxidant Capacities

#### 2.4.1. Analysis of Ferric-Reducing Activity Power (FRAP)

The FRAP was measured according to a previously reported method with slight modifications [20]. Briefly, 5 μL od AOP solution (0.25, 0.5, 1, 2, 4, and 8 mg/mL) was gently mixed with 180 μL of FRAP kit (Beyotime, S0116) for 10 min at room temperature. Then, the absorbance was read at 593 nm (Varioskan Flash, Thermo Scientific, USA) in accordance with the suggestions of the supplier. Values were calculated according to the calibration curve with aqueous solutions of FeSO_4_·7H_2_O in the range of 0–1800 μmol/L (y = 0.116x − 0.1087, R^2^ = 0.9929). The final results are expressed as the concentrations of FeSO_4_·7H_2_O with equivalent antioxidant activity.

#### 2.4.2. Analysis of ABTS Radical Scavenging Ability

BTS radical scavenging activities were evaluated according to a previously reported method with minor modifications [24]. Briefly, AOPs solutions (0.25, 0.5, 1, 2, 4, and 8 mg/mL) were gently blended with an ABTS kit (Beyotime, S0121) for 10 min at room temperature. Then, the absorbance was read at 734 nm (Varioskan Flash) in accordance with the suggestions of the supplier. The ABTS radical scavenging activity was calculated using the following equation:$$ABTS\;radical\;scavenging\;activity\left(\%\right)=[\frac{A_{ABTS}-A_{\mathrm{blank}}}{A_{ABTS}}]\times 100$$
where A_ABTS_ is the absorbance of the ABTS radical solution without the sample, A_sample_ is the absorbance of the ABTS radical solution with the measured samples, and A_blank_ is the absorbance of the ABTS radical solution with distilled water.

### 2.5. Statistical Analysis

All data were recorded from three repeated experiments, and the results are expressed as means ± standard deviation. Statistical analysis was applied using Prism 10 (GraphPad Software, Inc., La Jolla, CA, USA). Comparisons between two or multiple groups were analyzed using Student’s *t*-test and one-way analysis of variance (ANOVA), respectively. *p* < 0.05 denoted a statistically significant difference. Design Expert 13.0.1 (Stat-Ease, Inc., Minneapolis, MN, USA) was used to establish the regression model, composing the Box–Behnken design and predicting the optimal conditions for the yield of AOPs from the ultrasonic-assisted extraction experiment.

## 3. Results

### 3.1. Optimization of Ultrasonic-Assisted Extraction Conditions on AOPs

Response surface methodology (RSM), an effective collection of statistical techniques, is commonly used to optimize the complex experimental parameters and their interactions [25]. BBD, an RSM-based statistical tool, is usually used to predict the relationship between the experimental result and calculated output results [26]. In this study, the BBD was used to optimize ultrasonic-assisted extraction parameters within the minimal number of trials. The BBD experimental data are summarized in Table 1; moreover, a second-order polynomial equation was generated to express the mathematical model using multiple regression analysis. The final equation is as follows:Y = 4.66 − 1.09A + 0.2831B + 0.3005C − 0.0228AB − 0.1586AC + 0.5352BC − 0.6554A^2^ − 0.1834B^2^ − 0.4954C^2^
where Y is the yield of AOP (%); A, B, and C represent ultrasonic power (W), ultrasonic time (min), and ratio of water to material (mL/g), respectively.

As shown in Table 2, the one-way analysis of variance was used to confirm whether there was a statistically significant difference among the effects of the ultrasonic parameters on the yield of AOPs. The results revealed that this model is significant, having a relatively high *F*-value (14.01) and an inferior *p*-value (0.0011), indicating that the probability of error in this model was less than 0.11%. Furthermore, the lack of fit was not significant, which is shown by an *F*-value of 5.28 and a *p*-value of 0.07, suggesting that this model was reliable because the value of the coefficient of variation (CV, 8.69%) was less than 10 and had a high value of adequate precision (adequate precision, 10.9830). In addition, the value of the coefficient of determination (0.9474) and the difference in value between the adjusted coefficient of determination (0.8798) and the predicted coefficient of determination (0.3115) were higher than 0.2. Simultaneously, the linear coefficients of the extraction parameters (A and C), interaction coefficient (BC), and quadratic term coefficients (A^2^ and C^2^) of this fitted model were significant, with a *p*-value lower than 0.05, while the interaction coefficients (AB and AC) were not significant, with a *p*-value higher than 0.05 (Table 2). These above results indicate that all selected extraction parameters could significantly affect the yield of AOPs.

In addition, the shapes of the contour plots represent the significant differences between mutual interactions. The circular contour plot indicates that the interactions are nonsignificant, whereas the elliptical contour demonstrates that the interactions are significant [27]. The three-dimensional response surface plots and two-dimensional contour plots of this fitted model are shown in Figure 1. It can be visually observed that the interaction effects between the ultrasonic time and the ratio of water to material are significant due to its elliptical shape (Figure 1f), which could also be proved via its *p*-value of the BC interaction coefficient (0.0229 < 0.05) and that ultrasonic power is the most significant factor affecting the yield of AOPs due to the lowest *p*-value (<0.0001).

#### Optimal Ultrasonic-Assisted Extraction Conditions and Their Validation

The suitability of the model’s quadratic equations for predicting the optimal response values was checked using the selected conditions. The optimal conditions were adjusted as follows: ultrasonic time, 6 min; ratio of water to material, 11.99 mL/g; and ultrasonic power, 382.53 W. Under these conditions, the predicted yield was 5.72%. However, considering the actual operability, the operated conditions were modified as follows: ultrasonic time, 6 min; ratio of water to material, 12 mL/g; and ultrasonic power, 380 W. Under these modified conditions, the yield of AOPs was 5.72 ± 0.07%, which closely agrees with the predicted yield and simultaneously reveals that this model is satisfactory and valid.

### 3.2. Physiochemical Properties and Structural Characteristics of AOPs

#### 3.2.1. Chemical Compositions of AOPs

The compositions of AOPs are displayed in Table 3. AOP30 shows a high content of polysaccharides (26.97%) and a low content of uronic acid (14.67%) among three fractions, indicating that AOPs are acidic polysaccharides. Interestingly, the content of protein is below 0.2%, demonstrating that AOPs might emerge as polysaccharide–protein complexes. 

#### 3.2.2. Molecular Weights, Monosaccharide Compositions, and Zeta Potential of AOPs

It is commonly thought that the molecular weights and monosaccharides of polysaccharides play an irreplaceable role due to their antioxidant activities. The monosaccharide compositions of AOPs were investigated using HPSEC chromatograms. As shown in Table 3, the molecular weights of AOP30 (11.07 kDa) were less than those of AOP50 and AOP70 (around 17 kDa), which might be due to its solubility [28]. Meanwhile, the polydispersity index (M_w_/M_n_) of AOP30, AOP50, and AOP70 was 3.23, 4.82, and 2.33, respectively, suggesting that AOP30 and AOP50 exhibit a relatively loose distribution [29]. A dissimilar result was obtained when the diameters of AOPs were different: that of AOP30 was nearly five to six times larger than AOP50 and AOP70 (Figure 2 and Table 3), respectively, indicating that AOP30 provided a loose distribution. The HPLC profiles of the monosaccharides released from the AOPs are shown in Figure 3a, and the results are presented in Table 3.

The types of monosaccharides found in AOPs were similar, which contained rhamnose, arabinose, and galactose. Notably, the AOPs were rich in glucose, especially AOP30 (89.88%). This result is consistent with that of a previous study [30], whereas the level of galactose in AOP50 and AOP70 was eight times higher than that found in AOP30. In addition, the level of mannose in AOP50 and AOP70 was double the level found in AOP30. However, galacturonic acid and glucuronic acid were not detected in AOP30, while they were detectable in AOP50 and AOP70. This result suggests that the AOPs were heteropolysaccharide fractions. Furthermore, AOP30 showed only one symmetric polysaccharide fraction (Figure 4), indicating that graded alcohol precipitation can be used to obtain relatively pure polysaccharide fraction from *A. officinarum*. The molecular weight of AOP30 (11.07 kDa) was much lower than that of AOP50 (17.40 kDa) and AOP70 (18.93 kDa); this result might be owing to the low content of galactomannan (the molar ratio of galactose and mannose in AOP30 was extremely lower than that in AOP50 and AOP70), which had a large molecular weight [31].

The Zeta potential is the electrical potential present at the hydrodynamic plane of shear that surrounds a charged particle. It essentially represents the potential at the specific point in space where low molecular weight ions cease moving with the particle and instead remain within the surrounding solvent [32]. Interestingly, the AOPs were negatively charged (Table 3), and AOP70 exhibited the lowest value within AOPs, which demonstrates that AOPs potentially donate electrons and are also in line with the contents of uronic acid. In addition, the size of AOPs (Figure 2) ranged from 526 to 2801 nm (Table 3).

#### 3.2.3. FTIR of AOPs

The FTIR spectra of AOPs ranging from 500 to 4000 cm^−1^ are shown in Figure 3b. A prominent, broad, and intense peak at 3421 cm^−1^ was observed, indicating the presence of the hydroxyl group. A weak band observed at 2929 cm^−1^ was assigned to the C − H stretching and bending vibration of C − H bonds. The absorption band centered at 1647 cm^−1^ was attributed to the asymmetric stretching vibration of C=O bonds. Additionally, the peaks at 1122 cm^−1^ suggested the presence of −O − C and C − O − H link bonds.

#### 3.2.4. Antioxidant Activities of AOPs

The scavenging ability of the ferric reducing power and the ABTS radical assay reveal the electron donating capacity [33], which have been widely applied for evaluating the antioxidant potential of polysaccharides [34,35]. In this study, AOPs increasingly reduced ferric ions, and AOP30 exhibited better reducing capacity than AOP50 and AOP70 (Figure 5a) at a concentration of 8 mg/mL. The reducing power of AOPs followed the order of AOP30 > AOP70 > AOP50. Conversely, AOPs demonstrated effective scavenging of ABTS radicals with increasing concentration (Figure 5b). The IC_50_ values of AOP30, AOP50, and AOP70 were 0.32, 0.63, and 0.17 mg/mL, respectively. The aforementioned results indicate that AOP30 may have a higher electron donating capability than AOP50, potentially due to its elevated levels of polysaccharides and uronic acid, as well as moderate polydispersity index (3.23). These characteristics suggest that AOP30 may exhibit a relatively wide distribution [29], which might provide more possibility for affinity between AOP30 and ABTS radicals. In addition, the Zeta potential of AOP30 was negatively charged, demonstrating that AOP30 might easily offer electrons.

## 4. Conclusions

Polysaccharides are regarded as significant bioactive macromolecules in *A*. *officinarum* Hance. However, there is still a limited understanding of the chemical structures and antioxidant activities of AOPs, which hinders their potential application in the functional food industry. Consequently, to investigate the potential application of AOPs, the optimization of the ultrasonic-assisted extraction of AOPs was conducted, and the structural properties, as well as antioxidant activities, were examined. The maximum extraction yield obtained was 5.72 ± 0.07%. When compared to the hot water method and the microwave-assisted method, the ultrasonic-assisted method required less time and water. Additionally, AOP30 demonstrated strong antioxidant activities. Importantly, AOP30 exhibited potent antioxidant activity, as indicated by its polydispersity index and negative Zeta potential. Overall, AOP30 shows excellent potential for development as a functional food ingredient in the food industry.

## Figures and Tables

**Figure 1 foods-13-00333-f001:**
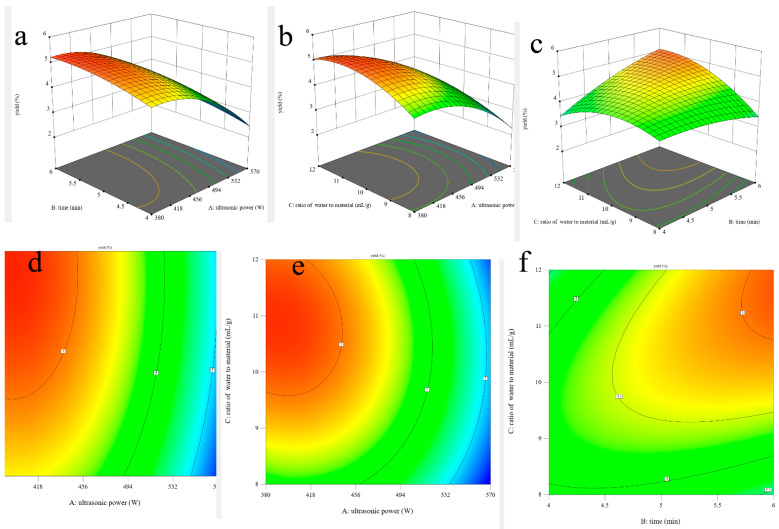
Three-dimensional surface plots (**a**–**c**) and two-dimensional contour plots (**d**–**f**) of ultrasonic-assisted extraction of AOPs.

**Figure 2 foods-13-00333-f002:**
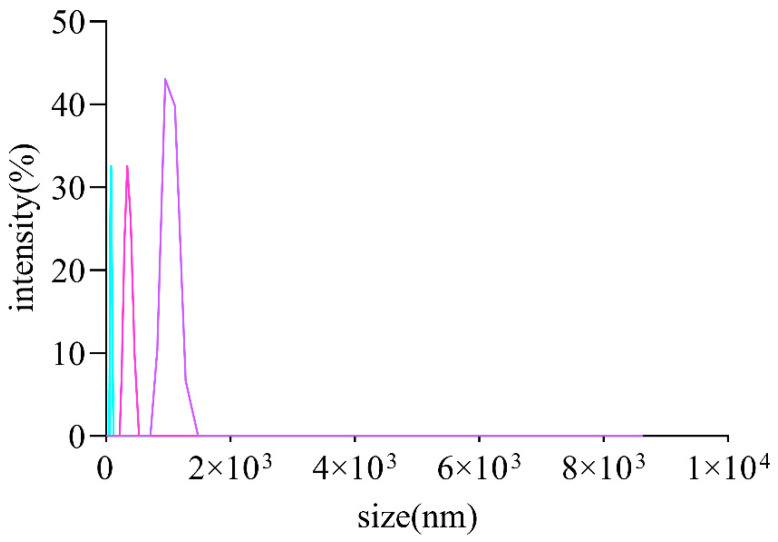
Size distribution curve of AOPs.

**Figure 3 foods-13-00333-f003:**
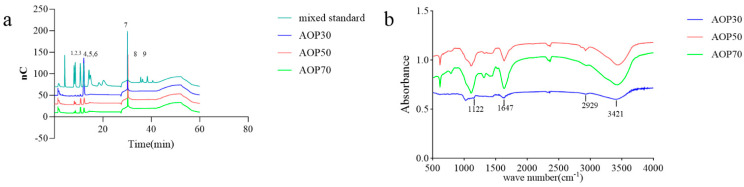
HPLC profiles of compositional monosaccharides (**a**), 1–9 indicates rhamnose, arabinose, galactose, glucose, xylose, mannose, solvent, galacturonic acid, glucuronic acid, glucuronic acid, respectively, and FT-IR spectra (**b**) of AOPs.

**Figure 4 foods-13-00333-f004:**
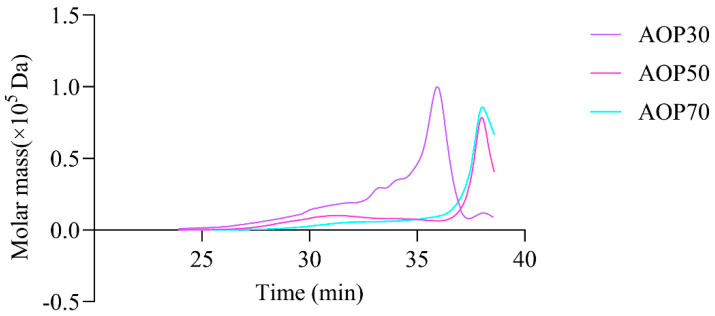
High performance size exclusion chromatography profiles of AOPs.

**Figure 5 foods-13-00333-f005:**
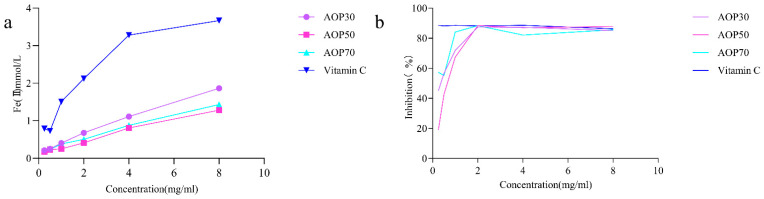
Antioxidant activities of AOPs: ferric reducing activity power (**a**); ABTS radical scavenging activity (**b**).

**Table 1 foods-13-00333-t001:** BBD with independent variables and observed values for the yield of AOPs.

Experiment	Levels of Extraction Parameters			Extraction Yield (%)
	A (W)	B (min)	C (mL/g)	
1	570	5	12	2.68
2	570	6	10	2.63
3	475	6	12	5.32
4	475	5	10	4.44
5	380	4	10	4.96
6	475	5	10	4.74
7	475	6	8	3.41
8	475	5	10	4.75
9	475	4	8	3.71
10	475	5	10	4.91
11	380	5	12	4.70
12	570	4	10	2.32
13	475	5	10	4.44
14	475	4	12	3.47
15	380	6	10	5.36
16	570	5	8	2.63
17	380	5	8	4.01

A: ultrasonic power (W); B: ultrasonic time (min); C: ratio of water to material (mL/g).

**Table 2 foods-13-00333-t002:** Analysis of variance for the fitted second-order polynomial model for the yield of AOPs.

Source	Sum of Squares	df	Mean Square	*F*-Value	*p*-Value
Model	15.45	9	1.72	14.01	0.0011 **
A	9.59	1	9.59	78.21	<0.0001 **
B	0.6413	1	0.6413	5.23	0.0560
C	0.7225	1	0.7225	5.89	0.0456 *
AB	0.0021	1	0.0021	0.0169	0.9002
AC	0.1006	1	0.1006	0.8211	0.3950
BC	1.15	1	1.15	9.35	0.0184
A^2^	1.81	1	1.81	14.75	0.0064
B^2^	0.1416	1	0.1416	1.16	0.3180
C^2^	1.03	1	1.03	8.43	0.0229
Residual	0.8581	7	0.1226		
Lack of fit	0.6849	3	0.2283	5.28	0.0710
Pure error	0.1731	4	0.0433		
Correlation total	16.31	16			

R^2^ = 0.9474, coefficient of variation (CV) = 8.69%, adjusted R^2^ = 0.8798, predicted R^2^ = 0.3115, and adequate precision = 10.9830; A—ultrasonic power (W); B—ultrasonic time (min); C—ratio of water to material (mL/g); * *p* < 0.05, ** *p* < 0.01.

**Table 3 foods-13-00333-t003:** Physicochemical properties, molecular mass distribution, and monosaccharide compositions of AOPs.

	AOP30	AOP50	AOP70
Polysaccharide (%)	26.97 ± 2.69 ^a^	14.05 ± 0.98 ^c^	19.47 ± 1.36 ^b^
Uronic acid (%)	14.67 ± 0.73 ^b^	11.56 ± 0.58 ^c^	17.89 ± 0.89 ^a^
Protein (%)	0.15 ± 0.06 ^b^	0.19 ± 0.08 ^a^	0.13 ± 0.05 ^c^
M_w(Da)_ × 10^3^	11.072	17.401	18.927
M_w_/M_n_	3.23	4.82	2.33
Diameter (nm)	2801.3 ± 98.61 ^a^	621 ± 60 ^c^	526 ± 30 ^b^
Zeta potential(mV)	−23.17 ± 0.71 ^b^	−27.3 ± 0.79 ^a^	−19.5 ± 3.67 ^c^
Monosaccharides (molar ratios, %)			
Rhamnose	0.26	16.56	6.92
Arabinose	2.08	11.18	13.39
Galactose	3.68	24.79	28.47
Glucose	89.88	24.83	24.03
Xylose	1.1	7.57	8.7
Mannose	3	6.49	8.79
Galacturonic acid	0	3.78	5.49
Glucuronic acid	0	4.22	3.62

Superscripts differ significantly (*p* < 0.05) among AOPs. Statistical significance was assessed using ANOVA followed by Duncan’s test.

## Data Availability

All of the data are contained within the article.

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
