# Peer review of "Ultrasonic-Assisted Extraction, Structural Characteristics, and Antioxidant Activities of Polysaccharides from Alpinia officinarum Hance"

_foods, 2024, doi:10.3390/foods13020333_

Round 1

Reviewer 1 Report

Comments and Suggestions for Authors

Dear authors,

   Thank you for your submission. The topic of finding a polysaccharide with antioxidant properties from herbs is impressive.

All methods in this research are sufficient in detail with appropriate statistical tests. However, some points need to be clear.

I would recommend a minor revision as follows:

Abstract: 

Please check the typographical errors and the redundancy of the sentences.

Materials and Methods:

  • Please rewrite the equations in this section. 
  • Lines 183-185: Please check the typographical errors.
  • The scientific name must be italicized.

Results:

Figure 2: Please adjust this figure as it is hard to read and follow. Besides Figure 2b, the authors can make a table to show the correlation between the peak in each wave number and chemical bonds.

Figure 3: This figure (a,b, and c) can be merged into one graph for easy to read and follow. 

Figure 4: This figure (a,b, and c) can be merged into one graph for easy to read and follow. 

Figure 5: The antioxidant value can be reported in the table. Besides, the IC50 value for ABTS should be calculated. 

Author Response

Abstract:

Please check the typographical errors and the redundancy of the sentences.

Response: Thanks for your suggestions, we have revised the words and sentences carefully.

Materials and Methods:

Please rewrite the equations in this section.

Response: Thanks for your suggestions, we have revised the equations.

Lines 183-185: Please check the typographical errors.

Response: thanks for your suggestions, we have revised this sentence.

The scientific name must be italicized.

 Response: thanks for your suggestions, we have revised the format of scientific name.

Results:

Figure 2: Please adjust this figure as it is hard to read and follow. Besides Figure 2b, the authors can make a table to show the correlation between the peak in each wave number and chemical bonds.

Response: thanks for your suggestions, we have adjusted figure2, which makes it clearly, on the other hand, only four characteristic absorption peaks were observed, hence, we directly marked them in Figure 2b.

Figure 3: This figure (a,b, and c) can be merged into one graph for easy to read and follow.

Response: thanks for your suggestions, we have revised them and merged one graph.

Figure 4: This figure (a,b, and c) can be merged into one graph for easy to read and follow.

Response: thanks for your suggestions, we have revised them and merged one graph.

Figure 5: The antioxidant value can be reported in the table. Besides, the IC50 value for ABTS should be calculated.

Response: thanks for your suggestions, when we compared data presentation forms of table and graph, it was found that line graph showed stronger visuality than table, thus, line graph was selected, moreover, we have calculated IC50 values of AOPs for ABTS radical scavenging ability, the IC50 values of AOP30, AOP50 and AOP70 were 0.32, 0.63 and 0.17 mg/mL, respectively.

Reviewer 2 Report

Comments and Suggestions for Authors

The present study aimed to optimize ultrasonic-assisted extraction parameters of AOP by Box-Behnken design, analyzed structural features, importantly, elucidated the underlying mechanism behind antioxidant activities of AOPs.

Lines 16, 60, 62, 68, 300: There are several typing errors in the whole manuscript, like .. impr0ove, 24mL/g, 20min, ofAOP, (table3), etc; check the whole manuscript, please.

Line 22-24: The result showed that the predicted optimal ultrasonic extraction parameters were as followed: ultrasonic time of 6 min; ratio of water to material of 12 mL/g; ultrasonic power of 380 W. Under these conditions, the maximal predicted yield of AOP was 5.72%,

Line 91: A. officinarum Hance rhizome was purchased from market… please include how many grams were purchased for the study, were they purchased in the same day or during different days or even better different seasons of the year?. The main concern of this study is that the amount of structural characteristics and AOP are affected by the stage of growth of the rhizome and for this study was not taken into account, therefore the optimization of ultrasonic-assisted extraction parameters of AOP by Box-Behnken design are particular of the evaluated samples.

Line 354: compared to the hot water and microwave-assisted methods reported in the literature, the AOP30 extraction method required less time and water.

Comments on the Quality of English Language

Minor editing of English language required

Author Response

The present study aimed to optimize ultrasonic-assisted extraction parameters of AOP by Box-Behnken design, analyzed structural features, importantly, elucidated the underlying mechanism behind antioxidant activities of AOPs.

  1. Lines 16, 60, 62, 68, 300: There are several typing errors in the whole manuscript, like .. impr0ove, 24mL/g, 20min, ofAOP, (table3), etc; check the whole manuscript, please.

Response: sorry for our mistake, we have revised them.

  1. Line 22-24: The result showed that the predicted optimal ultrasonic extraction parameters were as followed: ultrasonic time of 6 min; ratio of water to material of 12 mL/g; ultrasonic power of 380 W. Under these conditions, the maximal predicted yield of AOP was 5.72%,

Response: sorry for our mistake, we have revised them.

  1. Line 91: A. officinarum Hance rhizome was purchased from market please include how many grams were purchased for the study, were they purchased in the same day or during different days or even better different seasons of the year?. The main concern of this study is that the amount of structural characteristics and AOP are affected by the stage of growth of the rhizome and for this study was not taken into account, therefore the optimization of ultrasonic-assisted extraction parameters of AOP by Box-Behnken design are particular of the evaluated samples.

Response: thanks for your suggestions, in fact, the rhizome of A. officinarum Hance can be harvested every four years. Therefore, it is difficult to obtain different seasons of raw material. Besides, in this study, five kilograms of raw material were purchased from the local market.

  1. Line 354: compared to the hot water and microwave-assisted methods reported in the literature, the AOP30 extraction method required less time and water.

Response: sorry for our mistake, we have revised this sentence, it should be “When compared to the hot water method and microwave-assisted method, the ultrasonic-assisted method required less time and water”.

  1. Comments on the Quality of English Language

Minor editing of English language required.

Response: thanks for your suggestions, we have revised manuscript carefully.

Reviewer 3 Report

Comments and Suggestions for Authors

The manuscript focuses on the ultrasonic-assisted extraction and chemical characterization of polysaccharides extracted from Alpinia officinarum Hance , including its structural characteristics and antioxidant activities. While the basic idea of he study is interesting and could contribute to new insights into the use of such a spice for health in the future, the manuscript could benefit some improvements seemingly essential.

Major comments:

1. While the study focuses in the use of ultrasonic as a method for extracting polysaccharides in the spice, a deeper analysis of existing literature should be performed to determine whether ultrasonic is the most pertinent method to be applied. Have there been other studies comparing different methods of extraction? How could the authors be sure that ultrasonic was the most suitable method? This should be clearly stated in the manuscript.

2. The same concern goes to the use of water as the extraction liquid/medium. Has there been any evidence of the effectivity of other extraction media such as ethanol for polysaccharides? 

3. L. 127-128: the authors should explain the meaning of the fractions. What do the digits in AOP30, AOP50, and AOP70 mean?

4. Table 2: Please reconsider the inclusion of this table in the manuscript. I somehow think this table is not necessary.

5. Figure 1: require a better figure in a larger size. The image is somehow blurry.

6. Overall, the manuscript should be reviewed extensively for the use of English and formatting (spacing, the use of capital lettering, etc). The correction should include, but not limited to the following minor comments I have indicated.

Minor comments:

L. 16: A. officinarum should be italicized, typo: impr0ove

L. 40: foreign terminology (in Chinese) should be italicized

L. 66: manufacturing (with lowercase letter)

L. 131, 152, 263, 268, 279: Capital letter at the beginning of paragraphs

L. 161: A sentence should not be initiated by a number. Please write "five" instead of 5

L. 183-185, 313: different font size

L. 187: ABTS, not BTS

L. 283-284, 309-310: The names of simple sugars do not need capital letter at the beginning of the words

L. 348: Species name in Latin should be italicized

Comments on the Quality of English Language

Overall, the manuscript should be reviewed extensively for the use of English and formatting (spacing, the use of capital lettering, etc). The correction should include, but not limited to the following minor comments I have indicated. Please use a professional language editing service to enhance your manuscript.

Author Response

The manuscript focuses on the ultrasonic-assisted extraction and chemical characterization of polysaccharides extracted from Alpinia officinarum Hance , including its structural characteristics and antioxidant activities. While the basic idea of he study is interesting and could contribute to new insights into the use of such a spice for health in the future, the manuscript could benefit some improvements seemingly essential.

Major comments:

  1. While the study focuses in the use of ultrasonic as a method for extracting polysaccharides in the spice, a deeper analysis of existing literature should be performed to determine whether ultrasonic is the most pertinent method to be applied. Have there been other studies comparing different methods of extraction? How could the authors be sure that ultrasonic was the most suitable method? This should be clearly stated in the manuscript.

Response: thanks for your suggestions, we have discussed the papers of extracting polysaccharides from A. officinarum Hance(AOP), many trials were investigated, hot water method was applied to isolate AOP under the conditions of extraction temperature, 95℃, extraction time, 3h, ratio of water to material, 43 mL/g [1]; enzymolysis method was also conducted with the conditions as followed: ratio of water to material, 24 mL/g, enzymolysis time, 50.5 min[2] scavenging ABTS radical with the IC50 value of 4.33 mg/mL [3]; furthermore, microwave assisted method was investigated with the conditions of microwave time, 20 min, microwave power, 350 W, ratio of water to material, 45 mL/g [4]. It could be accordingly confirmed two questions on the basis of above extraction methods when extracted AOP, firstly, a large amount of time was consumed, which was unable to meet the standards of food Manufacturing; secondly, a great amount of water required because of high liquid-solid ratio, which would take considerable time and energy to concentrate extracting solution. Hence, there was a highly demanded to develop an effective technology to separate AOP.we have revised them in the manuscript.

  1. The same concern goes to the use of water as the extraction liquid/medium. Has there been any evidence of the effectivity of other extraction media such as ethanol for polysaccharides?

Response: thanks for your suggestions, commonly,during the process of extracting polysaccharides from A. officinarum Hance, water is used to dissolve polysaccharides, while alcohol is utilized to precipitate polysaccharides.

  1. L. 127-128: the authors should explain the meaning of the fractions. What do the digits in AOP30, AOP50, and AOP70 mean?

Response: thanks for your questions, AOP was precipitated with30%, 50%, and70% alcohol, respectively. Then, three fractions (AOP30, AOP50, AOP70) were collected.

  1. Table 2: Please reconsider the inclusion of this table in the manuscript. I somehow think this table is not necessary.

Response: thanks for your suggestions, table 2 contains coefficients that are used to validate the fitted second-order polynomial model for ultrasonic-assisted extraction of AOP.

  1. Figure 1: require a better figure in a larger size. The image is somehow blurry.

Response: thanks for your suggestions, we have revised figure1.

  1. Overall, the manuscript should be reviewed extensively for the use of English and formatting (spacing, the use of capital lettering, etc). The correction should include, but not limited to the following minor comments I have indicated.

Minor comments:

  1. 16: A. officinarum should be italicized, typo: impr0ove

Response: thanks for your suggestions, we have revised them.

  1. 40: foreign terminology (in Chinese) should be italicized

Response: thanks for your suggestions, we have revised it.

  1. 66: manufacturing (with lowercase letter)

Response: thanks for your suggestions, we have revised it.

  1. 131, 152, 263, 268, 279: Capital letter at the beginning of paragraphs

Response: thanks for your suggestions, we have revised them.

  1. 161: A sentence should not be initiated by a number. Please write "five" instead of 5

Response: thanks for your suggestions, we have revised it.

  1. 183-185, 313: different font size

Response: thanks for your suggestions, we have revised them.

  1. 187: ABTS, not BTS

Response: thanks for your suggestions, we have revised them.

  1. 283-284, 309-310: The names of simple sugars do not need capital letter at the beginning of the words

Response: thanks for your suggestions, we have revised them.

  1. 348: Species name in Latin should be italicized

Response: thanks for your suggestions, we have revised it.

Comments on the Quality of English Language

Overall, the manuscript should be reviewed extensively for the use of English and formatting (spacing, the use of capital lettering, etc). The correction should include, but not limited to the following minor comments I have indicated. Please use a professional language editing service to enhance your manuscript.

Response: thanks for your suggestions, we have carefully revised them.

References

  1. Yi, Z.; Weidong, W.; Yong, L.; Yuanyuan, Z.; Jing, G. Optimization of extraction process and antioxidant activities of polysaccharides from alpinia officinarum hance. Food Science 2014, 35, 126-131.
  2. Liu, Y.; Zhang, X.; Wang, Y.; Yang, G. Optimization for extraction process of polysaccharide from alpinia galanga willd. By enzymolysis method and its antioxidant activity analysis. Journal of Southern Agriculture 2016, 47, 1376-1382.
  3. Yi, W.; sisi, L.; honghe, Z. Optimization of extraction process of alpinia officinarum hance polysaccharides and study on antioxidant activity. Farm Products Processing 2023, 34-38.
  4. hui,, W.S.; miao,, H.M.; Ling, W.W.Z. Microwave extraction process of galangal polysaccharide. Journal of Wuhan Polytechnic University 2021, 40, 102-107.

Round 2

Reviewer 3 Report

Comments and Suggestions for Authors

Most of the comments have been addressed. However, please check the formatting of all figures and tables that somehow seems misplaced to me.